# Prevalence of overweight and its associated factors among Malaysian adults: Findings from a nationally representative survey

Chean Tat Chong[ID]*, Wai Kent Lai[ID], Syafinaz Mohd Sallehuddin, Shubash Shander Ganapathy

Institute for Public Health, National Institutes of Health, Ministry of Health Malaysia, Shah Alam, Selangor, Malaysia

* chean@moh.gov.my

**Data Availability Statement:** These are third party data. The data can be requested via National Institutes of Health - Data Repository System (NIH-DaRS), Ministry of Health Malaysia at https://

## Abstract

The World Health Organization has reported that the prevalence of overweight is a growing problem in many countries, including middle- and lower-income countries like Malaysia. This study aimed to determine the prevalence of overweight and its associated factors among Malaysian adults. A total of 9782 Malaysian adults aged 18 and above were included in this study, representing states and federal territories from the National Health and Morbidity Survey 2019. Sociodemographic data (sex, locality, age, marital status, ethnicity, educational level, income level, and health literacy), non-communicable disease status (hypertension, diabetes, and hypercholesterolemia), and lifestyle behaviours (physical activity level, smoking status, and also fruit and vegetable consumption) were collected and analysed to identify factors associated with overweight. The study found that the prevalence of overweight among Malaysian adults was 50.1%. Multivariate analyses showed that several factors, including female gender [aOR (95% CI) = 1.33 (1.11, 1.58); p = .002], ages 30–59 years [aOR (95% CI) = 1.61 (1.31, 1.97); p < .001], being Malay [aOR (95% CI) = 1.68 (1.36, 2.07); p < .001], Indian [aOR (95% CI) = 2.59 (1.80, 3.74); p < .001] or other Bumiputera [aOR (95% CI) = 1.82 (1.38, 2.39); p < .001], being married [aOR (95% CI) = 1.23 (1.00, 1.50); p = .046], and having adequate health literacy [aOR (95% CI) = 1.19 (1.01, 1.39); p = .033], were significantly associated with an increased risk of overweight. Additionally, overweight individuals had a significantly higher risk of non-communicable diseases such as diabetes [aOR (95% CI) = 1.47 (1.23, 1.75); p < .001] and hypertension [aOR (95% CI) = 2.60 (2.20, 3.07); p < .001]. The study suggests that intervention programs should be implemented in an equitable and cost-effective manner to target these high-risk populations and address the burden of overweight in Malaysia.

## Introduction

Body mass index (BMI) is a commonly utilized tool for evaluating the nutritional status of adults, and it is associated with a range of health risks [1, 2]. In the past decade, there haven't

nihdars.nih.gov.my/. Anyone will be able to access these data in the same manner as the authors and that the authors did not have any special access privileges that others would not have.

**Funding:** We appreciate the funding and support from the Ministry of Health Malaysia (NMRR-18-3085-44207, awarded to SSG. The funders had no role in study design, data collection and analysis, decision to publish, or preparation of the manuscript.

**Competing interests:** The authors have declared that no competing interests exist.

been many studies on global overweight and obesity trends. However, the studies conducted between 1980 and 2013 showed that the number of adults with a BMI of 25 or higher increased worldwide. For men, it went up from 28.8% to 36.9%, and for women, it rose from 29.8% to 38.0% [3]. The prevalence of obesity, defined as a BMI of 30 or higher, also increased significantly. In men, it went from 3.2% in 1975 to 10.8% in 2014, and in women, it went from 6.4% to 14.9% [4]. The number of obese men went from 31 million in 1975 to 281 million in 2016, while obese women increased from 69 million to 390 million in the same period. Additionally, there were 1.30 billion adults who were overweight but not obese [5]. The global prevalence of obesity has nearly tripled since 1975, affecting many countries, including those with low and moderate incomes [5]. In Malaysia, a nationwide survey conducted in 2019 revealed a rising trend in obesity prevalence, increasing from 15.1% in 2011, to 17.7% in 2015, and finally to 19.9% in 2019 [6–8].

The increasing prevalence of overweight and obesity has been linked to the development of non-communicable diseases (NCDs), including stroke, cardiovascular disease, and diabetes [2]. This trend has resulted in a significant global health burden [9], contributing to higher rates of disability-adjusted life years (DALYs) and mortality [10]. Recent studies have also suggested that overweight and obesity may compromise the immune system, thereby increasing the risk of infectious diseases [11]. As such, addressing overweight and obesity and promoting healthy weight management strategies is crucial to reduce the health burden associated with NCDs and infectious diseases [12].

Despite the implementation of various recommendations and intervention programs aimed at preventing overweight and obesity, research has shown that achieving and maintaining weight reduction can be challenging [13]. Therefore, the aim of this study was to determine the prevalence of overweight among Malaysian adults and its associated factors. The findings of this study will provide crucial information for prioritizing and developing precise planning guidelines to effectively combat the rising prevalence of overweight in Malaysia.

## Methods

### Study design and sampling

This study involved the secondary analysis of data collected from the National Health and Morbidity Survey (NHMS) 2019, which is a cross-sectional population-based survey conducted in Malaysia. The survey was conducted nationwide between July 14th and October 2nd, 2019, with an achievement response rate of 87.2% [8]. The study population included all adults aged 18 years and above who had lived in non-institutional living quarters (LQs) for at least two weeks in all states and federal territories of Malaysia. Individuals living in institutional living quarters, such as hotels, hostels, hospitals, and others, were not included in this study.

A two-stage stratified random sampling technique was used in this study to achieve national representativeness. The main stratum included all states and federal territories, while the secondary stratum consisted of both urban and rural strata from the main stratum. The main sampling unit involved 475 Enumeration Blocks (EBs), comprising 362 EBs from urban areas and 113 EBs from rural areas. The secondary sampling unit involved a total of 5,676 living quarters (LQs), comprising 4,320 LQs from urban areas and 1,356 LQs from rural areas within the selected EBs. The classification of urban and rural areas was defined by Department of Statistics Malaysia. The full details of the research methods and sampling design used in this nationwide survey are available in the official report [8].

## Data collection

The data collection process for this nationwide survey included face-to-face interviews and self-administered process using structured questionnaires programmed into application in tablet. For respondents who preferred paper-based questionnaires, they were also provided. Structured questionnaires were used to gather information on sociodemographic factors and lifestyle behaviours, while calibrated equipment was utilized to collect data on non-communicable disease (NCD) status, body weight, and height from respondents aged 18 and above.

## Independent variables

This study obtained sociodemographic data including sex, locality, age, marital status, ethnicity, educational level, income level, and health literacy, as well as NCD status (hypertension, diabetes, and hypercholesterolemia) and lifestyle behaviours (physical activity level, smoking status, and fruit and vegetable consumption) from the respondents. These data were analysed to determine the factors associated with overweight.

This study collected various sociodemographic data from the respondents. Age was categorized into three groups: 18 to 29 years, 30 to 59 years, and 60 years and above. Ethnicity was classified into five categories: Malays, Chinese, Indians, other Bumiputera (Sabah and Sarawak), and others. Respondents' income was grouped into four categories: < RM 1000, RM 1000–1999, RM 2000–2999, and ≥ RM 3000. Working status was classified into two categories: working and not working. Educational level was categorized as no formal education, primary education (completion of Standard Six), secondary school (completion of Form Five), and tertiary education (completion of Form Six, obtained certificate, diploma or any other academic degree). Marital status was divided into three groups: single, married, and other (divorced, and widow or widower). The respondents' health literacy was assessed using the HLS-M-Q18, an adapted version of HLS-EU-Q47 [14], where a score of 0 to 33 was classified as limited health literacy, and a score above 33 was classified as adequate health literacy.

The NCD status of the respondents was evaluated by determining the presence of diabetes, hypertension, and hypercholesterolemia through either self-reported diagnosis by healthcare professionals or clinical measurement using appropriate devices. The CardioChek portable blood test device was used to obtain finger-prick blood glucose (BG) and total cholesterol (TC) levels, with fasting capillary BG ≥ 7.0 mmol/L and/or non-fasting capillary BG ≥ 11.1 mmol/L indicating diabetes [15], and TC levels ≥ 5.2 mmol/L indicating hypercholesterolemia [16]. Blood pressure (BP) was measured using the HEM-907 BP Monitor (Kyoto, Japan), with systolic BP ≥ 140 mm Hg and/or diastolic BP ≥ 90 mm Hg indicating hypertension [17].

This study utilized validated questionnaires to determine smoking status and level of physical activity. The Global Adult Tobacco Surveillance (GATS) [18] questionnaire was used to identify current smoking status, while the short version of the International Physical Activity Questionnaire (IPAQ) [19] was used to assess the level of physical activity. The respondents' fruit and vegetable consumption were also collected and was considered adequate if they consumed a minimum of five servings daily, and inadequate if they consumed less than five servings daily. Criteria for active physical activity level were defined as a minimum of 20 minutes per day of three vigorous activities, or a minimum of 30 minutes per day of five days of moderate-intensity activity or walking, or a minimum of 600 MET-minutes per week of a combination of walking, moderate-intensity, or vigorous-intensity activities for at least five days, or a minimum of 1500 MET-minutes per week of three days of vigorous-intensity activity, or a minimum of 3000 MET-minutes per week of a combination of walking, moderate-intensity, or vigorous-intensity activities for at least seven days.

## Dependent variable

This study employed a standard protocol to measure body height and weight twice for each participant, with the final measurements calculated from the average of the two readings. Body height was determined using the SECA Stadiometer 213 (SECA GmbH & Co. KG, Germany) to the nearest 0.1 cm, while body weight was measured using the Tanita Personal Scale HD 319 (Tanita Corporation, Japan) to the nearest 0.1 kg. Prior to measurement, the devices were calibrated and participants were instructed to wear lightweight clothing, remove their shoes, and stand upright on the measuring equipment. Body mass index (BMI) was computed by dividing body weight in kilograms by the square of body height in meters ($kg/m^2$), where BMI values of 25.0–29.9 $kg/m^2$ and $\geq 30$ $kg/m^2$ were used to define overweight and obesity, respectively [20]. For the present analysis, only adults with a BMI $\geq 25.0$ $kg/m^2$ (inclusive of obesity) were included to examine the independent determinants of overweight.

## Statistical analysis

Statistical analysis was performed using IBM SPSS Statistics (version 28, Chicago, Illinois, USA). Descriptive statistics were used to summarize the data, and the Pearson's $\chi^2$ test was used to evaluate differences between BMI categories and independent variables. A multiple logistic regression model was employed to identify the factors associated with overweight, while controlling for potential confounders. All variables with a P-value less than .25 were included in the final model, with an adjusted odds ratio (AOR) obtained for each variable. A P-value less than .25 was considered sufficient for controlling residual confounding [21]. Both crude odds ratio (COR) and AOR were presented with 95% confidence intervals (CI), and P-values less than .05 were considered statistically significant.

## Ethical approval

This study obtained ethical approval from the Medical Research and Ethics Committee (MREC), Ministry of Health Malaysia. It was registered under the National Medical Research Registry (NMRR) with number NMRR-18-3085-44207. All participants provided written informed consent prior to interviews during the NHMS's 2019 data collection.

## Results

The study involved a total of 9782 respondents, predominantly comprising Malays (51.3%), followed by Chinese (20.7%), other Bumiputera (11.3%), other ethnic groups (10.9%), and Indians (5.9%). The male respondents (52.4%) slightly outnumbered the females (47.6%), with the majority of the participants being married (63.9%) and aged between 30 and 59 years (54.1%). Nearly half of the respondents had completed secondary education (49.6%) and had an income level of RM3000 or more (53.4%). The majority of respondents resided in urban areas (77.9%) and were employed, with adequate health literacy. Among the respondents, 18.3% had diabetes, 30.1% had hypertension, and 39.0% had hypercholesterolemia. Regarding lifestyle behaviours, most of the respondents were physically active (77.1%), non-smokers (76.8%), and did not consume the recommended daily servings of fruits and vegetables (94.7%).

In the present study, the prevalence of overweight among adults in Malaysia was reported to be 50.1%. The prevalence of overweight was significantly higher among females (compared to males; $p < 0.001$); aged 30 to 59 years old (compared to other age group; $p < 0.001$); being Indians (compared to other ethnicities; $p < 0.001$); married (compared to other marital status; $p < 0.001$); adequate health literacy level (compared to limited health literacy level; $p < 0.001$);

diagnosed with diabetes (compared to non-diabetes; p < 0.001); hypertension (compared to non-hypertensive group; p < 0.001); high cholesterol level (compared to normal cholesterol level; p < 0.001); non-smoker (compared to smoker; p < 0.001); and those who were physically active (compared to physically inactive; p = 0.043). Details for the prevalence of overweight and variables were shown in Table 1.

Multiple logistic regression analysis was conducted to identify the independent factors associated with overweight among Malaysian adults. The findings, presented in Table 2, reveal that being females (AOR: 1.33, 95%CI: 1.11, 1.58) increases the likelihood of overweight. Similarly, adults aged 30 to 59 years old (AOR: 1.61; 95%CI: 1.31, 1.97) are at higher risk of being overweight compared to those aged 18 to 29 years old. Ethnicity also plays a role, with Malays (AOR: 1.68, 95%CI: 1.36, 2.07), Indians (AOR: 2.59, 95%CI: 1.80, 3.74), and other Bumiputera (AOR: 1.82, 95%CI: 1.38,2.39) having a significantly higher risk of overweight than Chinese. Married individuals (AOR: 1.23; 95%CI: 1.00, 1.50) were also found to be more likely to be overweight compared to single individuals. Surprisingly, adults with adequate health literacy (AOR: 1.19; 95%CI: 1.01, 1.39) were also at a higher risk of overweight compared to those with limited health literacy. The study also revealed that respondents with hypertension (AOR: 2.60; 95%CI: 2.20, 3.07) and diabetes (AOR: 1.47; 95%CI: 1.23, 1.75) had a higher likelihood of being overweight than respondents without these conditions. However, no significant association was found between cholesterol level and overweight. Finally, there was no significant association reported between lifestyle behaviours, such as physical activity and smoking status, and overweight among adults in Malaysia.

## Discussion

This study aimed to identify the prevalence and factors associated with overweight among adults in Malaysia from a national representative study. The results showed that 50.1% of Malaysian adults had overweight, which is an increase from previous national surveys conducted in 2011 [6] and 2015 [7], where the rates were 44.5% and 47.7%, respectively. This suggests that there has been a notable increase in the prevalence of overweight among Malaysian adults. Furthermore, the prevalence of overweight in Malaysia was higher than that observed in southern China [22], and most Southeast Asian countries [23–29], except for Brunei [30], but lower than the rates reported in the United States [31]. In Malaysia, the increasing prevalence of overweight can be attributed to the effects of rapid urbanization, greater food accessibility, and sedentary lifestyles [13], as well as the result of the nutrition transition which encompasses changes in food and beverage consumption patterns, influenced by factors such as the food environment, lifestyle modifications, behavioural changes, and government policies [32]. This finding is concerning, as it indicates that a significant proportion of Malaysian adults are overweight and emphasizing the necessity to expand public health policies and programmes regarding nutrition recommendations in Malaysia.

Multivariable logistic regression analysis was conducted to examine the independent determinants associated with overweight risk, after adjusting for potential confounding variables. Females had a significantly higher risk of being overweight compared to males, which is consistent with the global trend [33, 34]. The risk of obesity is higher among females compared to males in low- and middle-income countries, whereas in high-income countries, the prevalence of obesity shifts towards males. The adverse impacts of female obesity on reproductive health have been well-established [35, 36]. Additionally, adults aged 30–39, 40–49, and 50–59 years old were significantly more likely to be overweight compared to those aged 18–29 years old, while there was no significant association for those aged 60 years old and above. This finding is consistent with previous studies conducted in other countries [26, 28, 37]. The increased

**Table 1. Prevalence of overweight among adults Malaysian.**

| Variable | Prevalence | | P value* |
|---|---|---|---|
| | Non-overweight, n (%) | Overweight, n (%) | |
| Overall | 4389 (49.9) | 5393 (50.1) | |
| Sociodemographic Data | | | |
| Sex | | | < .001 |
| • Male | 2223 (54.1) | 2308 (45.9) | |
| • Female | 2166 (45.3) | 3085 (54.7) | |
| Age | | | < .001 |
| • 18–29 | 1253 (62.7) | 854 (37.3) | |
| • 30–59 | 2084 (42.1) | 3416 (57.9) | |
| • ≥ 60 | 1052 (50.8) | 1123 (49.2) | |
| Locality | | | 0.704 |
| • Rural | 1704 (50.5) | 2105 (49.5) | |
| • Urban | 2685 (49.7) | 3288 (50.3) | |
| Ethnicity | | | < .001 |
| • Malays | 2728 (46.4) | 3577 (53.6) | |
| • Chinese | 665 (59.7) | 540 (40.3) | |
| • Indians | 228 (36.3) | 395 (63.7) | |
| • Other Bumiputera | 460 (44.8) | 594 (55.2) | |
| • Others | 308 (60.4) | 287 (39.6) | |
| Educational Level | | | 0.184 |
| • No Formal Education | 285 (56.8) | 271 (43.2) | |
| • Primary Education | 974 (51.1) | 1211 (48.9) | |
| • Secondary Education | 2073 (48.7) | 2642 (51.3) | |
| • Tertiary Education | 1046 (49.9) | 1260 (50.1) | |
| Marital Status | | | < .001 |
| • Single | 1209 (61.2) | 881 (38.8) | |
| • Married | 2733 (44.9) | 3948 (55.1) | |
| • Others | 447 (47.9) | 564 (52.1) | |
| Working Status | | | 0.095 |
| • Working | 2615 (50.9) | 3147 (49.1) | |
| • Not Working | 1774 (48.1) | 2246 (51.9) | |
| Income Level | | | 0.451 |
| • Less than RM 1000 | 480 (53.6) | 477 (46.4) | |
| • RM 1000 –RM 1999 | 902 (51.7) | 1068 (48.3) | |
| • RM 2000 –RM 2999 | 706 (49.9) | 930 (50.1) | |
| • RM 3000 & above | 2062 (49.2) | 2597 (50.8) | |
| Health Literacy | | | **0.001** |
| • Limited | 1511 (54.1) | 1638 (45.9) | |
| • Adequate | 2491 (47.7) | 3300 (52.3) | |
| NCD Status | | | |
| Diabetes | | | < .001 |
| • No | 3586 (53.1) | 3762 (46.9) | |
| • Yes | 803 (35.8) | 1628 (64.2) | |
| Hypertension | | | < .001 |
| • No | 3159 (57.0) | 2847 (43.0) | |
| • Yes | 1229 (33.5) | 2543 (66.5) | |

(*Continued*)

**Table 1.** (Continued)

| Variable | Prevalence | | P value* |
| --- | --- | --- | --- |
| | Non-overweight, n (%) | Overweight, n (%) | |
| Hypercholesterolemia | | | < .001 |
| • No | 2616 (55.2) | 2608 (44.8) | |
| • Yes | 1773 (41.5) | 2785 (58.5) | |
| Lifestyle Behaviour | | | |
| Smoking Status | | | < .001 |
| • No | 3329 (48.1) | 4472 (51.9) | |
| • Yes | 1043 (55.8) | 905 (44.2) | |
| Physical Activity | | | **.043** |
| • Inactive | 1070 (52.7) | 1214 (47.3) | |
| • Active | 3274 (49.2) | 4127 (50.8) | |
| Fruit & Vegetable Intake | | | .225 |
| • Inadequate | 3573 (49.0) | 4506 (51.0) | |
| • Adequate | 153 (54.7) | 202 (45.3) | |

*P values of less than .05 were considered significant.

financial independence, easy access to food, and availability of calorie-dense food in lower- and middle-income countries may contribute to the higher prevalence of overweight among adults [38].

The present study conducted a comparison of the ethnicity of adults, revealing a higher risk of overweight among Malays, Indians, and other Bumiputera in contrast to Chinese. This finding was consistent with previous national population surveys [13] and a study conducted in Singapore [39]. The observed differences in dietary patterns across ethnic groups might explain this trend [40]. Additionally, the study found that married individuals were more likely to be overweight and obese compared to single people, a finding consistent with other studies [23, 25, 28, 41]. The hypothesis suggests that the likelihood of becoming obese is increased when one has an obese spouse, primarily due to shared lifestyle behaviors [42], including changes in diet and physical activity following marriage that contribute to the risk of obesity [43]. Interestingly, the study found that individuals with adequate health literacy were associated with a higher risk of overweight compared to those with limited health literacy. This finding is different from other studies [44, 45], although one study did not find an association between health literacy and overweight [46]. The variations in study outcomes regarding the association between health literacy and nutritional status may be attributed to the utilization of different instruments to measure health literacy, while acknowledging that this relationship can be influenced by additional factors such as lifestyle, geographical disparities, and cultural variations. Despite this finding, health literacy remains crucial in weight management [44, 45].

Also, the present study found that the higher risk of overweight was significantly associated with an increased likelihood of NCD, such as diabetes and hypertension. This finding is consistent with a pooled analysis of 1.8 million people from 97 prospective cohorts and approximately half of the risk of cardiovascular disease and three-quarters of the risk of stroke related to BMI were due to NCD [47]. Weight control measures have been established as indispensable and effective strategies in the prevention and progression of NCD, underscoring their significance in mitigating the global health impact of these conditions [48]. Therefore, proper planning and effective strategies are crucial to intervene and address the burden of disease related to overweight [9, 47–50].

**Table 2. Factors associated with overweight among adults Malaysians.**

| Variable | COR [95% CI] | P value[a] | AOR [95% CI]* | P value[b] |
|---|---|---|---|---|
| Sociodemographic Data | | | | |
| Sex | | | | |
| • Male | Ref | | Ref | |
| • Female | 1.42 (1.26,1.61) | < .001 | 1.33 (1.11,1.58) | **.002** |
| Age | | | | |
| • 18–29 | Ref | | Ref | |
| • 30–59 | 2.31 (1.98,2.70) | < .001 | 1.61 (1.31,1.97) | < **.001** |
| • ≥ 60 | 1.63 (1.34,1.98) | < .001 | 0.75 (0.54,1.04) | .088 |
| Ethnicity | | | | |
| • Malays | 1.71 (1.39,2.11) | < .001 | 1.68 (1.36,2.07) | < **.001** |
| • Indians | 2.60 (1.84,3.67) | < .001 | 2.59 (1.80,3.74) | < **.001** |
| • Other Bumiputra | 1.83 (1.40,2.38) | < .001 | 1.82 (1.38,2.39) | < **.001** |
| • Others | 0.97 (0.69,1.35) | .852 | 1.15 (0.81,1.62) | .434 |
| • Chinese | Ref | | Ref | |
| Marital Status | | | | |
| • Single | Ref | | Ref | |
| • Married | 1.93 (1.67,2.23) | < .001 | 1.23 (1.00,1.50) | **.046** |
| • Others | 1.72 (1.38,2.13) | < .001 | 0.92 (0.67,1.25) | .581 |
| Health Literacy | | | | |
| • Limited | Ref | | Ref | |
| • Adequate | 1.29 (1.11,1.51) | .001 | 1.19 (1.01,1.39) | **.033** |
| NCD Status | | | | |
| Diabetes | | | | |
| • No | Ref | | Ref | |
| • Yes | 2.03 (1.73,2.73) | < .001 | 1.47 (1.23,1.75) | < **.001** |
| Hypertension | | | | |
| • No | Ref | | Ref | |
| • Yes | 2.63 (2.30,3.01) | < .001 | 2.60 (2.20,3.07) | < **.001** |
| Hypercholesterolemia | | | | |
| • No | Ref | | Ref | |
| • Yes | 1.74 (1.52,2.00) | < .001 | 1.16 (0.99,1.35) | .062 |
| Lifestyle Behaviour | | | | |
| Smoking Status | | | | |
| • No | Ref | | Ref | |
| • Yes | 0.74 (0.63,0.87) | < .001 | 0.85 (0.70,1.05) | .125 |
| Physical Activity | | | | |
| • Inactive | 0.87 (0.76,0.99) | .043 | 0.89 (0.77,1.03) | .120 |
| • Active | Ref | | Ref | |

Abbreviations: CI, Confidence Interval; COR, Crude Odds Ratio; AOR, Adjusted Odds Ratio.

[a]P value of less than .25 were included in the final model.

[b]P values of less than .05 were considered significant.

*Multiple logistic regression was applied. The final model was adjusted for sex, age, ethnicity, marital status, health literacy, diabetes, hypertension, hypercholesterolemia, smoking status, and physical activity.

This study had several strengths, including the use of a large nationally representative population, which allowed for valid inferences to be made regarding Malaysian adults. Additionally, the use of validated and standardized instruments during data collection decreased the

likelihood of measurement inaccuracies. However, there were some limitations to this study, including its cross-sectional design, which prevented causal relationships to be established between exposure risk and disease outcomes. Furthermore, the inadequate dietary data collected in this study may have made it difficult to substantiate the association between food consumption and overweight among the population. Longitudinal studies are needed to further investigate the causal relationship between risk factors and overweight, as well as to evaluate the effectiveness of interventions to reduce the burden of overweight in Malaysia.

## Conclusion

The present study findings indicate that a high prevalence of overweight exists among adults in Malaysia, affecting approximately half of the population. The likelihood of being overweight and obese was associated with various factors, such as female gender, aged 30 to 59 years, being of Malay, Indian or other Bumiputera ethnicity, being married, and having adequate health literacy. Additionally, adults diagnosed with NCD (hypertension and diabetes) were more likely to be overweight. To effectively address the burden of overweight, public health nutrition interventions in Malaysia should focus on these high-risk populations using a cost-effective and equitable approach.

## Acknowledgments

The authors would like to thank the Director General of Health Malaysia for his permission to publish this paper. We would also like to thank all research team members and data collectors for their contributions and commitment in this study. We are also grateful for the kind cooperation of all participants.

## Author Contributions

**Conceptualization:** Chean Tat Chong.

**Data curation:** Syafinaz Mohd Sallehuddin.

**Formal analysis:** Wai Kent Lai.

**Funding acquisition:** Shubash Shander Ganapathy.

**Investigation:** Chean Tat Chong.

**Methodology:** Wai Kent Lai.

**Project administration:** Syafinaz Mohd Sallehuddin.

**Resources:** Syafinaz Mohd Sallehuddin.

**Supervision:** Chean Tat Chong.

**Validation:** Wai Kent Lai.

**Visualization:** Chean Tat Chong.

**Writing – original draft:** Chean Tat Chong.

**Writing – review & editing:** Chean Tat Chong.

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
