## [Decision Letter · Decision Letter 0]

18 Apr 2023

PONE-D-23-06209Prevalence of overweight and its associated factors among Malaysian adults: findings from a nationally representative surveyPLOS ONE

Dear Dr. CHONG,

Thank you for submitting your manuscript to PLOS ONE. After careful consideration, we feel that it has merit but does not fully meet PLOS ONE’s publication criteria as it currently stands. Therefore, we invite you to submit a revised version of the manuscript that addresses ALL the points raised during the review process.

We look forward to receiving your revised manuscript.

Kind regards,

Sofi G Julien, MD-PhD

Academic Editor

PLOS ONE

Journal Requirements:

Reviewers' comments:

Reviewer's Responses to Questions

**Comments to the Author**

1. Is the manuscript technically sound, and do the data support the conclusions?

Reviewer #1: Partly

Reviewer #2: Partly

2. Has the statistical analysis been performed appropriately and rigorously? 

Reviewer #1: Yes

Reviewer #2: Yes

3. Have the authors made all data underlying the findings in their manuscript fully available?

Reviewer #1: Yes

Reviewer #2: No

4. Is the manuscript presented in an intelligible fashion and written in standard English?

Reviewer #1: Yes

Reviewer #2: Yes

5. Review Comments to the Author

Reviewer #1: Dear authors:

It was a pleasure reading your manuscript. I have some comments. I look forward thou receive your edits.

1- Please enrich your introduction with more references from literature, with reference to prevalence of obesity worldwide.

2-in the questionnaire section please include a reference for the questions used. Which instrument was used?

thank you

Reviewer #2: Prevalence of overweight and its associated factors among Malaysian adults: findings from a nationally representative survey, written by Chong Chean Tat et al. on an interesting total sample of 9782 Malaysian adults aged 18 and above (better to mention the specific age range of the sample studied).

The aim of this preliminary cross-sectional study was to as determined determine the prevalence of overweight among Malaysian adults and its associated factors in all states and federal territories of Malaysia.

Age was categorized into three groups: 18 to 29 years, 30 to 59 years, and 60 years and above with multivariable logistic regression analysis conducted to examine the independent determinants associated with overweight risk, after adjusting for potential confounding variables.

Firstly it is safe to say that the English writing is fluid and easy to read and that the study is interesting and provides further preliminary insight in the overweight inducing factors among Malaysian adults.

Nevertheless, the abstract should describe numerically the main results that are summarized while in the results section to add the specific p value next to the respective ranges of odd ratios discussed.

The methodology describes well the steps undertaken to proceed with the collation of data.

In the discussion, saying that the results are similar or on the contrary contrast with previous studies should be expanded to clarify more how.

Further research is required to better understand the key preliminary data analyzed by the authors on the broad subject of overweight and obesity.

6. PLOS authors have the option to publish the peer review history of their article (what does this mean?). If published, this will include your full peer review and any attached files.

Reviewer #1: **Yes: **Mireille SERHAN

Reviewer #2: **Yes: **Marie Hokayem (Department of Nutrition and Food Sciences, Faculty of Arts and Sciences, Holy Spirit University of Kaslik (USEK), P.O. Box 446, Jounieh, Lebanon)

---

## [Author Response · Author response to Decision Letter 0]

23 May 2023

Dear Editor, 

Thank you for giving us the opportunity to revise the submitted manuscript into a better version. We have resubmitted all the revised documents in PLOS ONE's submission system for your further consideration. The authors received no specific funding for this work. The data were not available to the public. However, it can be obtained upon reasonable request and approval for access to the data at https://ebiostatistics.nih.gov.my/. Thank you for your consideration of this manuscript. 

Dear Reviewers,

We want to express our sincere appreciation for reviewing our manuscript. Your feedback and suggestions have been incredibly valuable in improving the quality of my research and we have amended it accordingly. Thank you for your time and expertise in providing a thorough and insightful review.

---

## [Editor Report · Decision Letter 1]

5 Jul 2023

Prevalence of overweight and its associated factors among Malaysian adults: findings from a nationally representative survey

PONE-D-23-06209R1

Dear Dr. CHONG,

We’re pleased to inform you that your manuscript has been judged scientifically suitable for publication and will be formally accepted for publication once it meets all outstanding technical requirements.

Kind regards,

Sofi G Julien, PhD

Academic Editor

PLOS ONE
---

## [Editor Report · Acceptance letter]

25 Jul 2023

PONE-D-23-06209R1 

Prevalence of overweight and its associated factors among Malaysian adults: findings from a nationally representative survey 

Dear Dr. Chong:

I'm pleased to inform you that your manuscript has been deemed suitable for publication in PLOS ONE. Congratulations! Your manuscript is now with our production department. 

Kind regards, 

on behalf of

Prof Sofi G Julien 

Academic Editor

PLOS ONE